# Physicochemical Characterization of Novel Epoxidized Vegetable Oil from Chia Seed Oil

**DOI:** 10.3390/ma15093250

**Published:** 2022-04-30

**Authors:** Ivan Dominguez-Candela, Alejandro Lerma-Canto, Salvador Cayetano Cardona, Jaime Lora, Vicent Fombuena

**Affiliations:** 1Instituto de Seguridad Industrial, Radiofísica y Medioambiental (ISIRYM), Universitat Politècnica de València (UPV), Plaza Ferrándiz y Carbonell, s/n, 03801 Alcoy, Spain; ivdocan@doctor.upv.es (I.D.-C.); scardona@iqn.upv.es (S.C.C.); jlora@iqn.upv.es (J.L.); 2Technological Institute of Materials (ITM), Universitat Politècnica de València (UPV), Plaza Ferrándiz y Carbonell 1, 03801 Alcoy, Spain; allercan@epsa.upv.es

**Keywords:** chia seed oil, fatty acids composition, epoxidized vegetable oil, epoxy equivalent weight

## Abstract

In this study, a novel epoxidized vegetable oil (EVO) from chia seed oil (CSO) has been obtained, with the aim to be employed in a great variety of green products related to the polymeric industry, as plasticizers and compatibilizers. Previous to the epoxidation process characterization, the fatty acid (FA) composition of CSO was analyzed using gas chromatography (GC). Epoxidation of CSO has been performed using peracetic acid formed in situ with hydrogen peroxide and acetic acid, applying sulfuric acid as catalyst. The effects of key parameters as temperature (60, 70, and 75 °C), the molar ratio of hydrogen peroxide:double bond (H_2_O_2_:DB) (0.75:1.0 and 1.50:1.0), and reaction time (0–8 h) were evaluated to obtain the highest relative oxirane oxygen yield (*Y_oo_*). The evaluation of the epoxidation process was carried out through iodine value (*IV*), oxirane oxygen content (*O_o_*), epoxy equivalent weight (*EEW*), and selectivity (*S*). The main functional groups were identified by means of FTIR and ^1^H NMR spectroscopy. Physical properties were compared in the different assays. The study of different parameters showed that the best epoxidation conditions were carried out at 75 °C and H_2_O_2_:DB (1.50:1), obtaining an *O_o_* value of 8.26% and an *EEW* of 193 (g·eq^−1^). These high values, even higher than those obtained for commercial epoxidized oils such as soybean or linseed oil, show the potential of the chemical modification of chia seed oil to be used in the development of biopolymers.

## 1. Introduction

Nowadays, fossil-based materials use is increasing greenhouse gas emissions, wastes in landfills, and the exhaustion of non-renewable resources [1]. This situation leads to the need to find new alternatives in order to decrease the elevated production of fossil-based materials. One of the most promising renewables resources is vegetable oils (VO) because of their availability, relatively low cost, and non-toxicity [2]. According to the latest data of European Bioplastics Association, the land used to produce the renewable feedstock is approximately 0.0013% in 2021 and is estimated to increase up to 0.058% in 2026 [3]. This increase continues to be a very low value compared with food and feed land used (25% in 2021), indicating that there is not competition between the renewable feedstock for feed, food, and the production of bioplastics. In case of VO production, it increases each year where approximately 20% is used for industrial applications due to the concern of environmental problems [4]. VO is mainly formed by triglycerides, which are composed of three fatty acids (FAs) connected to glycerol molecules. In their structure, carbon-carbon double bonds enable VO to be easily transformed, increasing their reactivity. One of the several ways to take advantage of these double bonds in VO is through the epoxidation reaction, which introduces oxirane oxygen in double bonds [5]. Epoxidized vegetable oils (EVOs) are used as reagents and intermediates in the manufacture of polymers and are commonly used as plasticizers, compatibilizers, stabilizers, reactive diluents, or epoxy matrices for composites [6,7,8,9]. Several EVOs have been reported in the literature, highlighting studies of linseed [10], cottonseed [11], soybean [12], karanja [13] or castor oil [14], among others [15,16,17,18].

Notwithstanding, one of the VO with the most significant potential, due to its high amount of double bonds present in the FA chains, is the chia seed oil (CSO) (*Salvia hispanica,* L.). As it is possible to observe in Table 1, the iodine value (*IV*) of CSO, which is a parameter used to determine the number of unsaturation in fats, oils, and waxes, is above 190 g I_2_/100 g. For this reason, CSO is one of the VO with the greatest potential for use in different sectors. Nowadays, CSO presents a market share of 20% of chia market with an expected increase of 23.4% from 2019 to 2025 in order to use it for food and non-food applications [19]. Regarding food applications, although is not widely used as others edible VOs, it is employed as food complements and cooking oil [20]. Some authors have also suggested the addition of CSO in food applications due to its high benefits [21,22]. On the other hand, as non-food applications, CSO has been epoxidized and employed as environmentally friendly plasticizer in one of the most promising industrial applications [23]. However, this study showed a lack of assessed parameters such as temperature and reagents concentration that can enhanced the epoxidation yield and no physico-chemical characterizations were carried out. Therefore, CSO could be one of the best VO candidates to be epoxidized, due to its high availability of double bonds, which can lead to obtain one of the highest theoretical oxirane oxygen content (*O_o_*) even higher than EVOs of greater industrial use (linseed and soybean). CSO has the potential to obtain more *O_o_* using fewer amounts of reagents (sulphuric and acetic acid or hydrogen peroxide e.g.), resulting in a low-cost process and a more environmentally friendly EVO [24,25,26].

Regarding the epoxidation process, selectivity (*S*) and *O_o_* vary depending on the catalyst used to obtain epoxides. Different methods have been studied, each with its advantages and disadvantages. For example, Guenter et al. employed molecular oxygen to carry out the epoxidation process [33]. This method could be low-cost and ecological using silver as the catalyst, but it is restricted to precursors like ethylene or butadiene and is not very efficient. The same authors studied the possibility of using halohydrins through hypohalous acids and their respective salts, but this route was not environmentally friendly [33]. The most employed route to synthesize EVOs is the use of hydrogen peroxide with in situ formed peroxyacid, as can be observed in Figure 1. This route presents some benefits compared to preformed peroxyacids, such as safer processing and handling, as well as requiring a minimum quantity of reagents to produce EVOs [34]. However, this route can lead to thermal runaway due to the exothermic reaction [35]. In order to reduce this risk, several actions can be considered such as comprehensive temperature control with adequate cooling capacity or dosing rate [36]. In this regard, this issue is taking into account industrially due to this route is the most employed [37]. Moreover, it is commonly known that, in many epoxidation processes, the excess of molar ratio reagents and/or higher temperatures can lead to side-reactions, being the most common the oxirane ring cleavage. The ring-opening has been studied for the purpose of comprehending the influence of factors in these side-reactions. For instance, Cai et al. [38] have studied the ring-opening for the epoxidation of cottonseed oil, where evaluated the effect of reagents taking into account the kinetic model. Furthermore, in the recent literature, the use of different epoxidation methods such as chemoenzymatic, polyoxometalates or heterogeneous catalytic systems in the presence of titanium silicate or ion exchange resin has been reported to overcome this drawback [4]. However, homogeneous catalysts have been proved to be more effective for industrial-scale plants to produce EVOs in batch reaction over 8 h [39,40]. In this regard, the use of strong mineral acid with acetic acid leads to a higher reaction rate [41].

The main purpose of this study was to develop a new EVO from chia seed oil (ECSO) as a raw material due to its great potential, determined by the high IV. This new EVO was carried out by in situ epoxidation process with acetic acid and hydrogen peroxide. In this way, a new reagent for use in the polymeric industry can be obtained through a naturally renewable resource as an alternative to the current petrochemical compounds or commercially available EVOs. After the analysis by GC of the composition of the fatty acids of chia oil obtained by cold mechanical extrusion, the epoxidation process was carried out, optimizing key parameters such as temperature, the molar ratio of H_2_O_2_:DB, and reaction time. The oxirane oxygen content (*O_o_*) was analyzed by titration method and the functional groups were confirmed using FTIR as well as ^1^H NMR spectroscopy. Further characterization was performed to evaluate the changes in physical-chemical parameters such as specific gravity, dynamic viscosity, and colorimetric coordinates. 

## 2. Materials and Methods

### 2.1. Materials

Chia seed (Salvia hispanica, L.) was supplied by Frutoseco (Bigastro, Alicante, Spain) and CSO was obtained by cold extraction using a CZR-309 press machine (Changyouxin Trading Co., Zhucheng, China). Aqueous hydrogen peroxide (30 wt.%), glacial acetic acid (99.7 wt.%), and sulphuric acid (96 wt.%) were purchased from Sigma Aldrich (Sigma Aldrich, Madrid, Spain). Reagents required for *IV* characterization such as Wijs solution (ICl), 0.1 N sodium thiosulphate solution (Na_2_S_2_O_3_), potassium iodide (KI), and cyclohexane (C_6_H_12_), were supplied from Sigma Aldrich (Sigma Aldrich, Madrid, Spain). To determine oxirane oxygen content, chlorobenzene (C_6_H_5_Cl), crystal violet indicator, potassium acid phthalate (KHC_8_H_4_O_4_), and 0.1 N HBr solution in glacial acetic acid were supplied from Panreac Química S.L.U. (Castellar del Vallés, Barcelona, Spain).

### 2.2. Epoxidation Process

Epoxidation reaction was performed in a three-necked round-bottom flask (1000 mL capacity). The flask, equipped with a two-bladed stirrer, was immersed in a thermostatic water bath where the temperature could be controlled to ± 0.1 °C of the desired temperature. A propeller mechanical stirrer was connected to the central neck, a drop-wise device was introduced in one of the side necks, and a thermometer was connected to the third neck. 

Epoxidation of CSO was carried out using a peroxyacid generated in situ from an organic acid. The most widely used in this process are formic (HCOOH) and acetic (CH_3_COOH) acids due to their high reactivity. The efficiency of these oxidants depends on the operating temperature, taking into account that according to literature, acetic acid is the most effective above 60 °C [42]. Furthermore, acting as the catalyst, sulphuric (H_2_SO_4_) and nitric acid (HNO_3_) are the most employed. In this regard, sulphuric acid with an optimal value of 3 wt.% with respect to the sum of the masses of hydrogen peroxide and acetic acid, was the most effective inorganic acid in order to obtain the highest conversion to oxirane oxygen, as was reported [43]. Therefore, the procedure is summarized as follows: firstly, 232 g of CSO (0.776 mol double bond for each 100 g of oil) and 25.3 g of glacial acetic acid were maintained at a constant temperature (60 °C) and stirring rate (220 rpm), which allowed a sufficient agitation to ensure proper mass transfer. After 10 min, sulphuric acid and hydrogen peroxide were mixed and added dropwise. The addition was completed within half an hour, at a constant rate following the method reported by Dinda et al. [44]. Two different H_2_O_2_:DB molar ratios (MR) mentioned previously (MR 0.75 and MR 1.50) at 60 °C were studied. MR 1.50 has been selected as the optimal amount of peroxide, which means 1.50 moles of H_2_O_2_ for one mole of double blonds [45]. It should be noted that CSO contains a higher iodine value compared to other VOs and, thus, a higher *O_o_* can be obtained. Therefore, MR 0.75 (half of the MR 1.50) has also been tested in this epoxidation process in order to study the effect of hydrogen peroxide and the possibility to achieve a reduction of the chemical compounds used. With respect to the temperature, different tests have been carried out at 60, 70, and 75 °C ± 1 °C, studying the reaction from 0 to 8 h considering the ranges reported in previous literature [46]. Samples were extracted at 0.5, 1.0, 1.5, 2.0, 4.0, 6.0 and 8.0 h, after adding the peroxide and sulphuric acid to monitor the reaction process. The collected samples were cooled at room temperature and then washed with water until they were acid-free up to pH neutral. To purify and remove the excess of water, samples were centrifuged 10 min at 7000 rpm.

### 2.3. Characterization

#### 2.3.1. Fatty Acid Composition

Prior to quantitative determination, FAs of oils have been transformed into methyl esters (FAMEs) following the standard method ISO 12966. The FAs profile was determined using HP/Agilent 6890 N gas chromatograph (Palo Alto, CA, USA) with Agilent 5973 N mass spectrometer detector (GC-MS). Equipment is provided with a splitless injector, ionization detector mode (70 eV), and integrator. The GC-MS method applied was according to ISO 12966. A DB-5MS capillary column (30 m length, 0.25 mm inner diameter, and 0.25 mm film thickness) from Teknokroma (Barcelona, Spain) was used using a temperature program from 140 to 300 °C at 20 °C/min. The injector and detector temperatures used were 280 and 300 °C, respectively. The sample injection volume was 1 µL using a splitless injector. Helium was used as carrier gas at a flow rate of 1 µL/min. Each FA was identified through the retention time comparison between pure commercial standard and the studied sample. FA results were expressed as percentages of the total regarding FAMEs.

#### 2.3.2. Iodine Value (IV)

The iodine value is referred to the mass of Iodine (I_2_ in grams) absorbed by 100 g of oil. It is an index used to determine the amount of unsaturation (double bonds) in FAs. This value was determined using Wijs solution (ICl) that reacted with double bonds and it was possible to detect the evolution by using sodium thiosulfate solution. *IV* was determined according to ISO 3961 using Equation (1):(1)$$IV=\frac{12.69\times c\times \left(V_{1}-V_{2}\right)}{m}$$
where IV refers to the mass of Iodine per 100 g of oil (g I_2_/100 g oil), c to the normality of sodium thiosulfate solution 0.1 N (Na_2_S_2_O_3_), $V_{1}$ to the volume of Na_2_S_2_O_3_ needed for titration of the blank (mL), $V_{2}$ to the volume of Na_2_S_2_O_3_ needed for titration of the sample (CSO) (mL), and m refers to the amount of sample used (g). Moreover, it is possible to determine the conversion of double bond ($X_{IV}$) using Equation (2) [10]:(2)$$X_{IV}\left(\%\right)=\left(\frac{IV_{0}-IV_{f}}{IV_{0}}\right)\times 100$$
where $IV_{0}$ is the initial iodine value of the VO sample and $IV_{f}$ is the final iodine value of the EVO after the epoxidation process. At least five measurements were made for each sample and the average values were reported.

#### 2.3.3. Oxirane Oxygen Content (*O_o_*)

The number of epoxy groups was determined using the direct method of titration with hydrobromic acid (HBr) solution in glacial acetic acid, using ASTM D1652. The sample was dissolved in chlorobenzene, followed by the addition of drops of crystal violet and titration using 0.1 N HBr in glacial acetic acid. The oxirane oxygen content (*O_o_*) was calculated with Equation (3):(3)$$O_{o}\left(\mathrm{wt}.\;\%\right)=1.6\times N_{i}\times \frac{\left(V-B\right)}{W}$$
where $N_{i}$ refers to the normality of HBr in glacial acetic acid, V to the volume of HBr solution for titration of the sample (mL), B to the volume of HBr solution for titration of the blank (mL), and W refers to the amount of sample used (g). At least five measurements were made for each sample and the average values were reported.

To determine the percentage conversion to oxirane ($Y_{OO}$) Equation (4) has been employed:(4)$$Y_{OO}\left(\%\right)=\left(\frac{O_{o}}{O_{the}}\right)\times 100$$
where, $O_{o}$ is the oxirane oxygen content experimentally obtained and $O_{the}$ is the theoretical maximum oxirane oxygen content that was calculated using Equation (5) [43]:(5)$$O_{the}\left(\mathrm{wt}.\;\%\right)=\left(\frac{\frac{IV_{o}}{2AM_{i}}}{\left\lceil 100+\left(\frac{IV_{o}}{2AM_{i}}\right)\times AM_{o}\right\rceil }\right)\times AM_{o}\times 100$$

Regarding Equation (5), $IV_{o}$ is the initial iodine value of the sample, $AM_{i}$ is the atomic mass of Iodine (126.9 g/mol), and $AM_{o}$ is the atomic mass of oxygen (16 g/mol). Moreover, the selectivity for oxirane oxygen (S) can be determined using Equation (6) [47].
(6)$$S=\left(\frac{O}{O_{the}}\right)\times \left(\frac{IV_{o}}{IV_{o}-IV_{f}}\right)$$

#### 2.3.4. Epoxy Equivalent Weight (EEW)

The epoxy equivalent weight (*EEW*) is defined as the mass, expressed in grams, of the epoxy resin which contains one equivalent of the epoxy group (g·eq^−1^). It is one of the most important features of epoxy resins, which is related to the crosslinking density and allows the calculation of the required amount of crosslinking agent for the curing process [48]. The *EEW* of ECSO was obtained following ASTM D1652 via titration using Equation (7):(7)$$EEW\;\left(g\cdot {\mathrm{eq}}^{-1}\right)=\left(\frac{1000\times W}{\left(V-B\right)\times N_{i}}\right)$$

#### 2.3.5. Fourier Transform InfraRed (FTIR) Spectroscopy

The substitution of double bonds and changes in functional groups was identified by Fourier Transform Infrared (FTIR) spectroscopy equipped with a horizontal attenuated total reflection modulus (ATR). Both CSO and ECSO were analyzed using a Bruker Vector 22 (Bruker Española, S. A, Madrid, España), averaging 20 scans at 4000–400 cm^−1^ and 4 cm^−1^ of resolution. It must be remarked that this method is immediate and straightforward to evaluate the possible change of the main functional groups, allowing to correlate the progress of epoxidations from infrared studies. However, the titration method obtains a better accuracy than the FTIR spectroscopy method, as has been reported by [15,49]. Thus, the FTIR method was used to verify the evolution of the main functional groups.

#### 2.3.6. Nuclear Magnetic Resonance (NMR) Spectroscopy

^1^H NMR spectroscopy was employed to compare and confirm the chemical structure of CSO and ECSO. Samples were analyzed using a Bruker AMX 500 unit (Bruker BioSpin GmbH, Rheinstetten, Germany) at 25 °C. Samples of 40 mg were dissolved in 0.6 mL of deuterated chloroform (CDCl_3_), mixed for 10 s and transferred to 5 mm NMR tubes for data acquisition.

#### 2.3.7. Physico-Chemical Properties

The method used to determine changes in specific gravity was a pycnometer, according to ASTM D1963. This method uses a 25 mL pycnometer maintaining a constant temperature of 25 °C. All densities of liquids were obtained against water. The specific gravities of untreated CSO and epoxidized were measured using Equation (8):(8)$$\rho_{r}=\frac{W_{s}-W_{e}}{W_{w}-W_{e}}\;$$
where $W_{s}$ is the weight (g) of the sample in the pycnometer, $W_{e}$ is the weight (g) of the empty pycnometer, and $W_{w}$ is the weight (g) of water in the pycnometer. At least five measurements were obtained with a maximum deviation of 3 × 10^−3^.

Dynamic viscosities were obtained using two Cannon-Fenske viscosimeters of 300 and 450 mm with flow ranges from 5 × 10^−5^ to 2.5 × 10^−4^ m^2^·s^−1^ and from 5 × 10^−4^ to 2.5 × 10^−2^ m^2^·s^−1^, respectively, at 20 °C. The assay has been carried out following the guidelines of the ASTM D-445. Viscosimeter was introduced in a water bath monitoring the temperature with a precision of ± 0.1 °C. At least five measurements were measured with a maximum dynamic viscosity deviation of 1.02 mPa·s.

The colorimetric coordinates of CSO and ECSO were measured using a Hunter Lab Colorimeter (Colour Flex, Hunter Associates Inc., Reston, VA, USA). The instrument (45°/0° geometry, D45 optical sensor, 10° observer) was calibrated before the experiments with Black and White reference tiles, and Green tiles were used to verify the correct operation. The values of luminance (L*) (0–100) represent lightness, parameters a* and b* indicate the approach from green (negative) to red (positive), and from blue (negative) to yellow (positive), respectively. At least five measurements were made for each sample and the average values were reported. Color variation was evaluated by using Equation (9):(9)$$∆E=\sqrt{∆L^{2}+∆a^{2}+∆b^{2}}$$

## 3. Results

### 3.1. CSO Extraction and Fatty Acid Composition

CSO was extracted by double cold extraction in a press machine to avoid chemical changes in the FA composition caused by high temperatures and the use of chemical solvents. Firstly, whole seeds were pressed to obtain oil and cake. Then, the residual cake was pressed again in order to obtain a higher yield of oil extraction. At the end of the pressing process, oil was filtered and centrifuged at 4000 rpm to be cleaned, and then it was stored in a cold dark room. The yield of extracted CSO was increased from 20.4% for the first process to 24.5% adding up the oil obtained from the second press of the chia cake. This slow increase of extraction yield with the second press shows that no longer than two presses are considered economically feasible, as reported by Kasote et al. [50]. This total yield is in the same range that was reported by Ixtaina et al. [51], in which the production yield of CSO was 24.8%.

The FA composition obtained by GC and the comparison with results published by other authors are shown in Table 2. As it can be observed, CSO presents a low proportion of saturated fatty acids (SFAs), with a value of 10.7%, integrated by myristic (0.06%), palmitic (7.2%), stearic (2.88%), and arachidic (0.55%) acids. With respect to monounsaturated fatty acids (MUFAs), it is observed even a lower proportion (4.41%), with palmitoleic (0.09%) and oleic (4.32%) acids. The higher content is found in polyunsaturated fatty acids (PUFAs) with 84.9%, which provides the most CSO double bonds: Linoleic (15.8%), γ-Linolenic (0.41%), and α-Linolenic (68.6%). The main difference between γ-Linolenic and α- Linolenic is the position of the double bond, where α-Linolenic contains the double bond in the 3rd, 6th, and 9th carbon position with respect to methyl terminus, whereas γ-Linolenic in 6th, 9th and 12th position. The high amount of α-Linolenic acid in the FAs profile provides better reactivity due to the position of double bonds in the carbon chain. The double bonds closer to the methyl terminus in the 3rd position present more reactivity than the 6th and 9th as has been reported by Scala and Wool [52], where the kinetics of the epoxidation process of vegetable oil was studied. Furthermore, CSO is the seed oil with the highest α-Linolenic content known compared to other studies of vegetable oils [53]. The results of the FAs of CSO are in concordance with the results reported by other authors, as is gathered in Table 2. The slight difference is probably due to the seed origin that has influenced FA composition. Some authors have reported that the FA profile and its quantity depend on several environmental factors such as temperature, light, or soil type [54].

### 3.2. Effect of Molar Ratio H_2_O_2_:Double Bond in the Epoxidation Process

To investigate the effect of molar ratio H_2_O_2_:DB, two experiments were made at MR 0.75 and 1.50 applying a constant temperature of 60 °C. In these conditions, both *IV* and *O_o_* analyses were obtained to monitor the epoxidation reaction. As it is possible to observe in Figure 2, *IV*_0_ was 197 g I_2_/100 g of oil for CSO, which is in accordance with Imran et al. [56] and Timilsena et al. [57], where both obtained values of 193 and 204 g I_2_/100 g of oil, respectively. The plot representation shows that once the epoxidation process takes place, the *IV* decreases due to the reaction of double bonds, as reported by Campanella et al. [58] with soybean oil. In the assay carried out at MR 1.50, the *IV* decreases more sharply, showing a greater rate yield in the process of double bonds substitution. However, although the results using MR 1.5 have been better than those with MR 0.75, the authors do not consider it appropriate to increase the MR above 1.5, due to an excess of reagents can cause side-reactions as the oxirane ring cleavage. In addition, the slope of the *IV* curve changes at the first 4 h, being less pronounced with MR 0.75 due to less initial oxygen active donor in the reaction for the lower hydrogen peroxide ratio, which is strongly related to the reduction of IV. Therefore, using MR 1.50 at 60 °C, *IV* is reduced up to 80.1 g I_2_/100 g of oil after 8 h of reaction, reaching almost a value of 60% for *X_IV_.* However, lower values were obtained using MR 0.75, with 42.2% for *X_IV_* and 114 g I_2_/100 g of oil for *IV*.

Figure 3 shows the plot evolution of *O_o_* and *EEW*. Inversely proportional to *IV*, when MR increases, the presence of *O_o_* increases as well. As reaction time proceeds, significant differences were observed comparing both MR. It is known that hydrogen peroxide is an active oxygen donor in the reaction [27]. Thus, when MR is increased from 0.75 to 1.50, an increase of *O_o_* can be noticed. The reason for this result is the increase in peracetic acid formation due to a higher amount of hydrogen peroxide [10]. Peracetic acid acts as a vector of oxygen, causing a conversion improvement. Then, doubling the amount of hydrogen peroxide, an increase of 36.8% in *O_o_* from 4.48 wt.% to 6.13 wt.% was obtained at the end of the reaction. Taking Equation (5) into account, a theoretical maximum oxirane oxygen content (*O_the_*) of 11.05% was obtained, thus achieving a *Y_oo_* value of 55.6% for MR 1.50. This epoxidation yield value can be ascribed to the lower efficiency of acetic acid at 60 °C, when the most effective temperature is observed above 60 °C [42]. Results are gathered in Table 3.

Related to selectivity (*S*) (values also summarized in Table 3), it is possible to appreciate that this value slightly decreases at MR 1.50 compared to MR 0.75. These results demonstrate that there are double bonds that were not replaced by epoxy groups. Although *X_IV_* and *Y_oo_* for MR 1.50 are higher than for MR 0.75, the opposite occurs with *S*. It is known that higher MR leads to an increment of *Y_oo_* and the presence of hydrolysis reactions, i.e., epoxy ring cleavage [28]. For that reason, *S* for MR 1.50 is slightly lesser than for MR 0.75 due to more presence of side-reactions caused by a higher amount of hydrogen peroxide. This is another reason why the use of higher reagent ratios is not technically feasible, as the high reactivity of the fatty acids present in chia oil causes unwanted side reactions. Therefore, the highest *O_o_* and lowest *EEW* values were obtained for MR 1.50 with values of 6.13 wt.% and 260 g·eq^−1^, respectively. 

### 3.3. Effect of the Temperature in the Epoxidation Process

The influence of temperature on the epoxidation process was studied at 60 °C, 70 °C, and 75 °C with the best MR (1.50) previously detailed. In Figure 4, it is possible to observe an increase in *IV* conversion at higher temperatures. As happened previously, the slope of *IV* showed a sharper change at the first 4 h. This behavior can be ascribed to the higher reactivity of double bonds present in the 3^rd^ position of methyl terminus of fatty acid in α-Linolenic acid [52], which react faster at the initial time of the experiments. In contrast, the less reactive double bonds (6th and 9th) present in α-Linolenic acid, Linoleic, and Oleic acid take a longer time to react. Then, increasing the temperature from 60 °C to 75 °C, an *IV* of 13.1 g I_2_/100 g of oil was obtained with regard to 80.1 g I_2_/100 g of oil at 60 °C. Therefore, the temperature is shown as the key factor to increase the conversion of *IV*.

Regarding the formation of epoxy groups plotted in Figure 5, it increases as temperature increases, accelerating the kinetic of epoxidation to form oxirane oxygen [46]. In the same way as with *IV*, the *O_o_* slope is higher at the first 4 h of reaction whereas, after that time, this trend decreases. This reduction in reaction rate can be related to the decomposition of peracetic acid, acetic acid, and hydrogen peroxide along the time [59]. With the higher temperature (75 °C) studied, it is possible to obtain the highest value of *O_o_*, 8.26 wt.%, or lowest *EEW*, 193 g·eq^−1^, obtaining a *Y_oo_* of 74.8%. An increase of 15 °C from 60 °C to 75 °C contributes to an improvement of 25.7% for *O_o_.* It should be noted that higher temperatures and extended time, increase the *O_o_* but can also cause a greater oxirane cleavage rate [44]. As Campanella et al. [60] and Gan et al. [61] studied for temperatures higher than 75 °C in soybean and palm oil, respectively, the oxirane ring was destabilized, which slowed the growth of the conversion to epoxy groups, even reducing the oxirane oxygen formed. In this sense, all temperatures studied showed almost no conversion to epoxy groups from 6 h onwards, being more pronounced with the temperature of 75 °C. For this reason, the authors did not consider it appropriate to carry out the epoxidation process at higher temperatures.

Regarding the selectivity, *S*, gathered in Table 3, it is possible to compare that it decreases substantially as temperature increases. At 70 °C and 75 °C the best *O_o_* have been obtained, as well as the lowest *S*. Thus, *S* decreases as higher is the temperature, which indicates an increase in oxirane cleavage. Although *Y_oo_* was 74.8% with these conditions, an interesting wt.% of *O_o_* or *EEW* have obtained if they are compared to other reports. Dinda et al. [44] used the same epoxidation method with cottonseed oil, reaching 4.96 wt.% for *O_o_* with 80% for *Y_oo_*. Furthermore, Mungroo et al. [5] studied the epoxidation of Canola oil using ion exchange resin as the catalyst, where 6.13 wt.% for *O_o_* and 90% for *Y_oo_* could be obtained. With the current values obtained with ECSO (8.26 wt.% for *O_o_* or *EEW* 193 g·eq^−1^), it is possible to forecast that it could be an excellent EVO as an alternative to the commercially available VO. A comparative example is found in the study carried out by Samper et al. [62], where epoxidized soybean oil with *EWW* of 238 g·eq^−1^, is used to manufacture a composite laminate with engineering applications. In addition, another example is reported by Fombuena et al. [63], where green composites are manufactured using commercial epoxidized linseed oil with 8 wt.% as epoxy matrix. Therefore, the present ECSO contains equal or even higher wt.% of *O_o_* or lower *EEW* than the most commercially available EVOs.

### 3.4. FTIR Analysis

As an alternative to evaluating the characterization of the epoxidation reaction through titration, FTIR spectroscopy is shown as an efficient tool for determining the representative peaks. Figure 6 shows the spectrum of untreated CSO, taken as reference. The characteristic peaks of double bonds are associated with 3010 cm^−1^ (=CH(_v_)) due to stretching of cis-olefinic bonds, 1652 cm^−1^ (C=C(_v_)) caused by stretching of disubstituted cis-olefins, and 723 cm^−1^ (C=C_(cis-δ)_) due to the combination of out-of-plane deformation and rocking vibration in cis-disubstituted olefins. Other characteristics peaks associated to methyl and methylene groups are obtained from 2961 to 2851 cm^−1^ (-CH_3(asym-v)_ and -CH_2(sym and asym-ν)_) and from 1462 to 1375 cm^−1^ (-CH_2(asym-δ)_ and –CH_3(sym-δ)_). Finally, the peak at 1743 cm^−1^ represents a carbonyl stretching (C=O_(v)_) of ester groups, and also the peak obtained at 821 cm^−1^ (C-O-C_(v)_) should be noted, barely visible in the untreated CSO but indicative of oxirane oxygen. Compared to the characteristic peaks of untreated CSO described in the literature, such as Timilsena et al. [64], a slight difference is shown due to the origin of seed and climate conditions, which can influence the fatty acid composition.

The monitoring of the epoxidation reaction by FTIR spectroscopy at different MR and temperatures has been focused on the characteristic peaks corresponding to double bonds (Figure 7) and the plot evolution of the oxirane oxygen content (*O_o_*) (Figure 8). All spectra are obtained after 8 h of epoxidation reaction time. Figure 7a represents the plot evolution of the peak at 3010 cm^−1^ (=CH_(v)_). It is possible to observe that the peak decreases drastically using MR 1.50 at 75 °C, which is in concordance with the analysis done by titration. This indicates the low quantity of available doubles bonds after the epoxidation reaction. Figure 7b,c show the graphical evolution of the peaks at 1652 (C=C_(v)_) and 723 cm^−1^ (C=C_(cis-δ)_), respectively, with the same trend mentioned previously. In general, the characteristic peaks of double bonds decrease when MR increase from 0.75 to 1.50 and the temperature reaches 75 °C, obtaining fewer available double bonds.

On the other hand, the increase of MR from 0.75 to 1.50, leads to a higher intensity of the oxirane oxygen group peak, (C-O-C_(v)_), located at 821 cm^−1^, following the same trend as the titration method as it is observed in Figure 8a. This new molecular group, not detectable in CSO, increases due to the insertion of oxygen into the double bonds through peracetic acid formed by the epoxidation reaction process. In parallel form, in Figure 8b, the same trend for the hydroxyl group (-OH_(v)_) can be observed at 3470 cm^−1^. In this case, greater MR accelerates the presence of hydroxyl groups by oxirane ring decomposition in the ECSO structure, as has been reported by Goud et al. [65]. The main reason is that the great amount of hydrogen peroxide employed contributes to the epoxy group formation, whereas another quantity is led to opening the oxirane ring obtaining hydroxyl groups [45].

With regard to the temperature effect, the temperature at 75 °C increases the intensity of the peaks corresponding to the hydroxyl group, compared with the MR effect. It indicates that the opening of the oxirane ring is more pronounced at higher temperatures than the MR ranges studied. This behavior has also been reported by Dinda et al. [44] during cottonseed epoxidation, where the highest content of hydroxyl groups was obtained at 75 °C. In addition, this effect is increased with VOs with higher initial IV, which contributes to obtaining a greater epoxy conversion as well as hydroxyl groups formation in a simultaneous process [66].

### 3.5. ^1^H NMR Analysis

^1^H NMR spectra were obtained to confirm the change of functional groups in ECSO synthesized with the best conditions in the epoxidation process, i.e., MR 1.50 at 75 °C. In Figure 9 the ^1^H NMR spectra of CSO and ECSO are plotted, showing a signal intensity (A) at 5.3–5.5 ppm for CSO. This characteristic peak corresponds to vinyl hydrogens from double bonds, which almost disappeared for the ECSO sample due to the conversion of *IV*. This peak was split into two signals: on the one hand, a small signal of vinyl hydrogen from double bonds located at 5.6 ppm, indicating that few double bonds remain after epoxidation reaction as was corroborated by *IV* and FTIR; on the other hand, a signal at 5.3 ppm for the central hydrogen of the glyceride moiety [67]. In addition, two more signals were observed at 2.02 ppm (C), associated with hydrogen adjacent to double bonds, and at 2.8 ppm (B), corresponding to allyl hydrogen between double bonds. After the epoxidation process, the peak at 2.8 ppm disappeared completely to show a displacement to 1.5 ppm (F) assigned to methylene hydrogens adjacent to oxirane groups in ECSO. The newly formed group was also registered in two new peaks between 2.85–3.00 (E) and 3.00–3.25 ppm (D) in ECSO. The first one corresponds to methylene hydrogens between two oxirane oxygen groups, whereas the second is related to hydrogens of the carbons of the new oxirane oxygen group. The evidence of hydroxyl groups formed by the opening of the oxirane ring was observed in the 4–3.4 ppm region as was described by M. Farias et al. [68], who evaluated the epoxidation of soybean oil using a homogeneous catalyst such as molybdenum (IV) complex. In that study a lower conversion to oxirane oxygen and selectivity at 80 °C was reported compared to this paper. The ^1^H NMR spectra obtained were consistent with the previous analysis, confirming the formation of oxirane oxygen in double bonds of the ECSO sample as well as hydroxyl groups as a result of side-reactions.

### 3.6. Physico-Chemical Properties

Physico-chemical properties such as specific gravity, dynamic viscosity, and colorimetric coordinates have been measured before and after the epoxidation reaction process of the CSO at different MR and temperatures. These parameters can be taken as a quick and easy methodology to monitor the epoxidation process. Results obtained after 8 h of reaction time are gathered in Table 4. With respect to specific gravity, low values in VO could be associated with the presence of cis double bonds present in FA structure, which difficult the packaging of the molecular chains [69]. The specific gravity value obtained in CSO, 0.9285, is in concordance with previous studies performed by Uzunova et al. [70] with values reported of 0.9288. As the epoxidation reaction advances and greater MR and temperatures are employed to improve the reaction yield, the insertion of oxirane oxygen in the FA structural chain increases the specific gravity of ECSO. Values reach up to 1.026, which is 10.5% higher than untreated CSO. The epoxy groups contribute to increase molecular mass without significant change in its volume due to the variation from sp^2^ hybridization of double bond (C=C) to sp^3^ hybridization of a single bond. This change improved chain packing, even though the stress on the chains was increased by the insertion of epoxy groups [69]. 

Regarding the dynamic viscosity of untreated CSO, it depends on the presence of unsaturations in the FA molecular chains. The presence of carbon-carbon double bonds kinks the fatty acid chains, increasing the average distance between them [71]. Therefore, an oil such as untreated CSO with high proportions of linolenic acid, i.e., a high amount of double bonds, results in lower dynamic viscosity compared to more saturated vegetable oils with an average value of 32 mPa·s. In epoxidized samples, dynamic viscosity is substantially higher as the performance of the epoxidation reaction progresses, reaching values of 558 mPa·s. These higher dynamic viscosities are ascribed to the increase of molecular weight and polarity in the structure compared to CSO, thus becoming stronger the interaction between molecules [72]. In addition, this property could also be increased, especially at high temperatures, due to the opening of the oxirane ring to form hydroxyl, ketone, and carboxylic groups that increase the intermolecular bonding and, consequently, increase the viscosity [73].

Finally, the variation of the color as a consequence of the epoxidation process can be observed in Figure 10. The untreated CSO is characterized by a yellow color, attributed to the high value of b* and L* (31.5 and 75.4 respectively), which are in concordance with values reported by Timilsena et al. [57]. As the yield of the reaction progresses by increasing the MR and the temperature, the greater presence of epoxy groups and lower *IV* in ECSO leads to decrease the yellow color, as it is possible to quantify by analyzing the b* parameter (decrease from 31.5 in CSO to 4.50 in ECSO). Regarding a*, a slightly reddish color (a* > 0) was observed in epoxidized samples with lesser epoxy groups (<6.3 wt.%) and higher *IV* (>80 g I_2_/100 g of oil) such as MR 0.75 and 1.50 at 60 °C. As the epoxidation yield increases, a* decreases from 6.73 for ECSO MR 0.75 at 60 °C up to negative values such as −3.69 and −2.85 for MR 1.50 at 70 °C and 75 °C, respectively, characterized by a lightly green color. Alarcon et al. [74] reported pale-yellow color in both epoxidized Baru and Macaw vegetable oils with epoxy content of 5.98 wt.% and 5.39 wt.%, respectively, but with low *IV* content (<15 g I_2_/100 g of oil) regardless the initial color of virgin oil. Therefore, the change of color observed from yellow (CSO) to reddish or to pale-yellow of different epoxidized samples can be attributed to the final *IV* change, regardless of the epoxy content of the epoxidized sample. Besides, the brightness (L*) of ECSO slightly increases after each epoxidation process, where the highest value was obtained for MR 1.50 at 75 °C. The noticeable color change can be quantified by analysing the color change (∆E). Aguero et al. [75] reported that a value greater than 5 implies a change visible to the naked eye. In this case, an increase of noticeable color change has been observed for all epoxidation conditions (∆*E* > 5) compared to CSO, where no significant changes were observed between 70 °C and 75 °C.

## 4. Conclusions

This research work assesses the development of a novel EVO obtained from chia seed oil (CSO). Previously to the epoxidation process, the fatty acid profile of CSO analyzed by GC showed polyunsaturated fatty acids (PUFAs) up to 84.9%, highlighting a proportion of α-Linolenic acid higher than 68.6%. These results make CSO an excellent candidate to be an alternative to the current commercial epoxidized oils. Variables such as MR and temperature have been studied in the epoxidation process. Results obtained with a MR 1.50 provide more *O_o_* caused by more peracetic acid formed that contributes to epoxy group formation. In a second step, the effect of the temperature was evaluated at 60, 70, and 75 °C. Epoxidation reaction at 75 °C provided the highest *O_o_* (8.26 wt.%) and lowest *EEW* (193 g·eq^−1^) with conversion to oxirane of 74.8%, and the reduction of the iodine value from 197 g I_2_/100 g oil up to 13.1 g I_2_/100 g oil, with a conversion for double bonds of 93.4%. Although the conversion to oxirane is less than 90%, *O_o_* obtained is similar to or even higher than available commercial epoxidized vegetable oils. The use of higher ratios of reactants and temperatures has not been carried out due to the problems provided by side-reactions as the oxirane ring cleavage, which reduces the efficiency of the process. The main functional groups formed by the epoxidation process have been confirmed by FTIR and ^1^H NMR spectroscopy, where both were in concordance with tendencies observed in the titration method. The influence of the epoxidation process in physico-chemical parameters such as specific gravity, dynamic viscosity, and color measurement has been reported, demonstrating that these parameters, as well as FTIR studies, can be used for rapid monitoring of the reaction. Results indicate that the epoxidation process implied a substantial increase in specific gravity and dynamic viscosity. All epoxidized samples showed an evident color variation (∆*E* > 5) from yellow (CSO) to light reddish for samples with higher *IV* (>80 g I_2_/100 g of oil) or to pale-yellow for samples with lower *IV* (<35 g I_2_/100 g of oil), that can be ascribed to the final *IV* change of epoxidized samples. Therefore, it is possible to conclude that ECSO is a highly potential EVO considered an alternative to the current epoxidized oils on the market, from an environmental point of view, and with high interest to be used in the manufacture of polymers and biopolymers.

## Figures and Tables

**Figure 1 materials-15-03250-f001:**
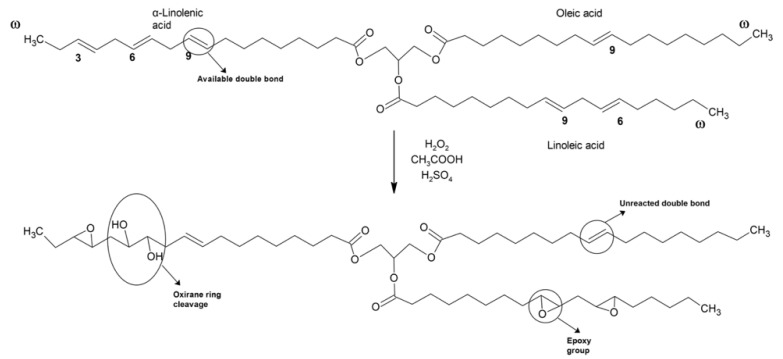
Schematic representation of epoxidation by means of in situ formed peroxyacids.

**Figure 2 materials-15-03250-f002:**
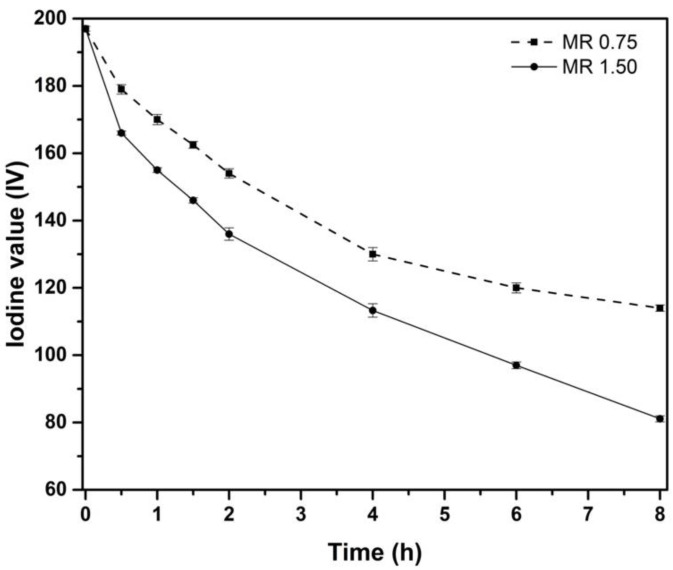
Effect on Iodine value (*IV*) of the MR 0.75 and 1.50 during the epoxidation process.

**Figure 3 materials-15-03250-f003:**
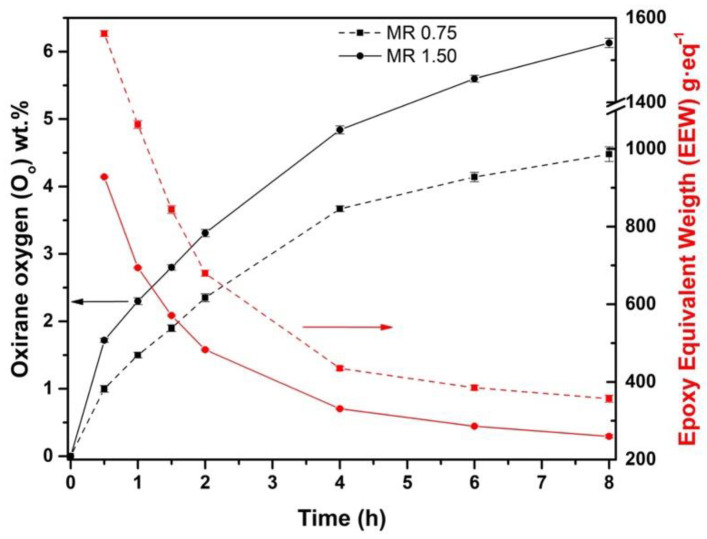
Effect on oxirane oxygen (*O_o_*) of the MR 0.75 and 1.50 at 60°C during the epoxidation process.

**Figure 4 materials-15-03250-f004:**
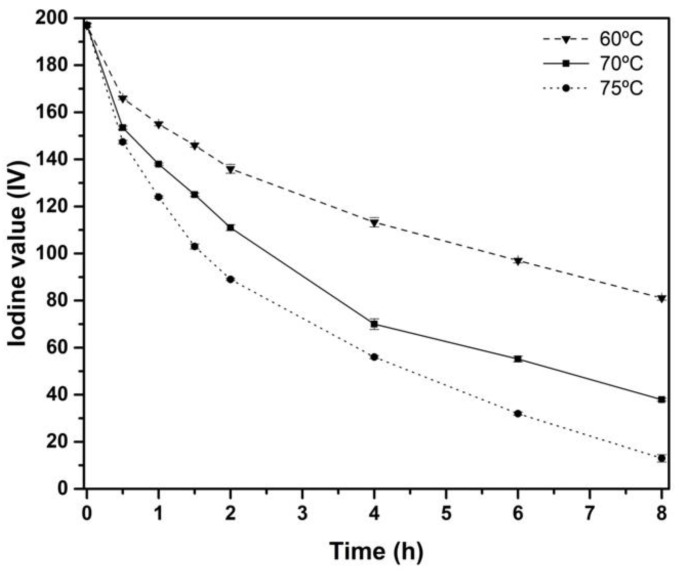
Effect on Iodine value (*IV*) of the temperature (60 °C, 70 °C, and 75 °C) with MR 1.50 during the epoxidation process.

**Figure 5 materials-15-03250-f005:**
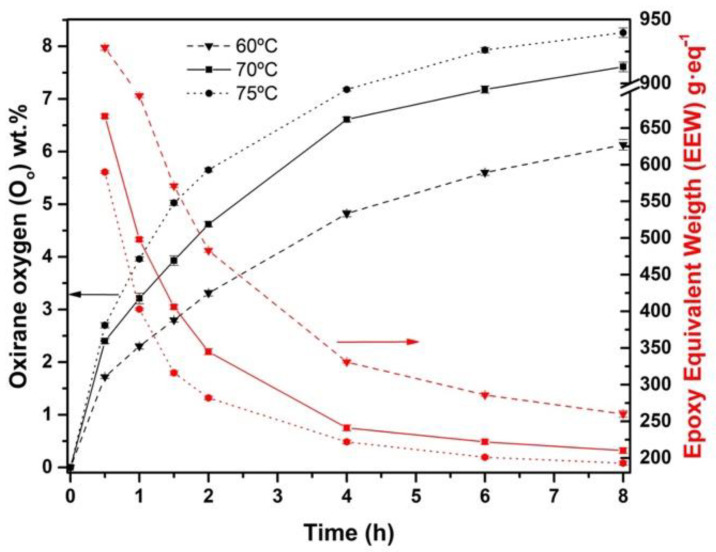
Effect on oxirane oxygen (*O_o_*) of the temperature (60 °C, 70 °C, and 75 °C) using MR 1.50 during the epoxidation process.

**Figure 6 materials-15-03250-f006:**
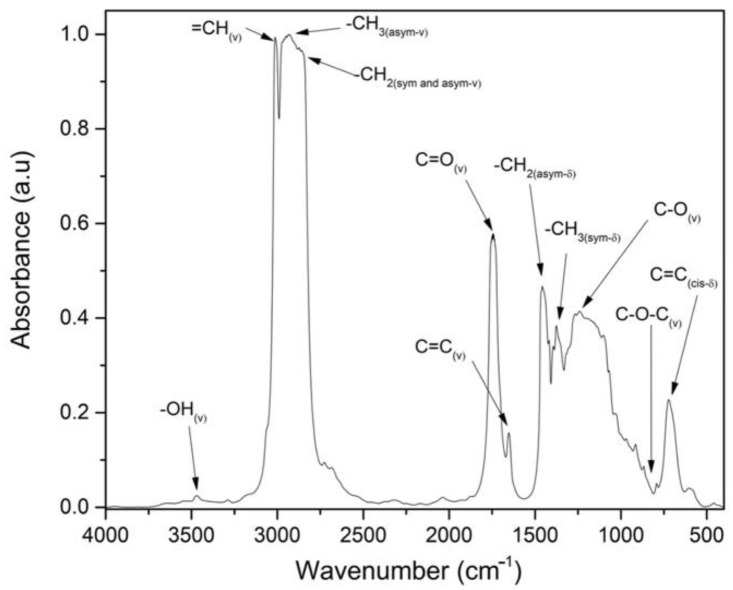
FTIR spectrum of CSO.

**Figure 7 materials-15-03250-f007:**
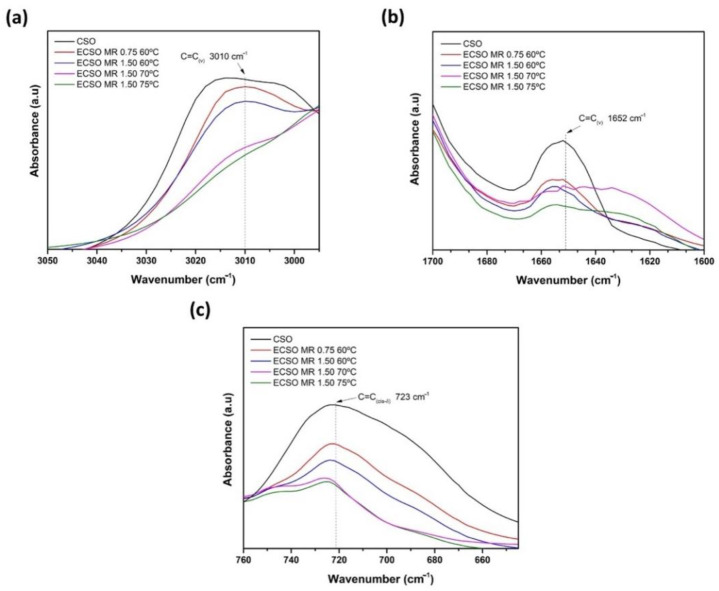
FTIR spectra of ECSO obtained with different epoxidation conditions by the analysis of characteristic peaks of the double bonds (**a**) 3010 cm^−1^ (=CH_(v)_), (**b**) 1652 cm^−1^ (C=C_(v)_) and (**c**) 723 cm^−1^ (C=C_(cis-δ)_).

**Figure 8 materials-15-03250-f008:**
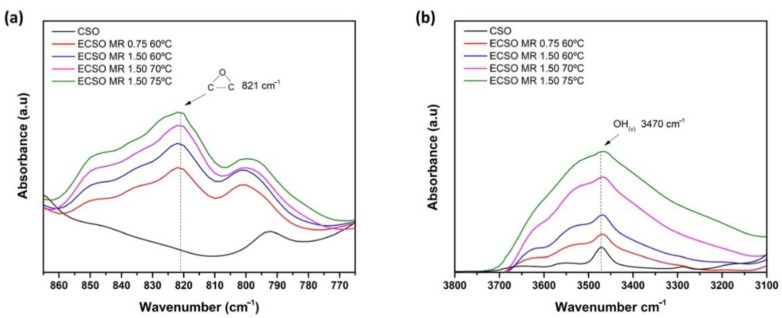
FTIR spectra of ECSO obtained with different epoxidation conditions by the analysis of characteristics peaks of (**a**) oxirane oxygen, and (**b**) hydroxyl groups.

**Figure 9 materials-15-03250-f009:**
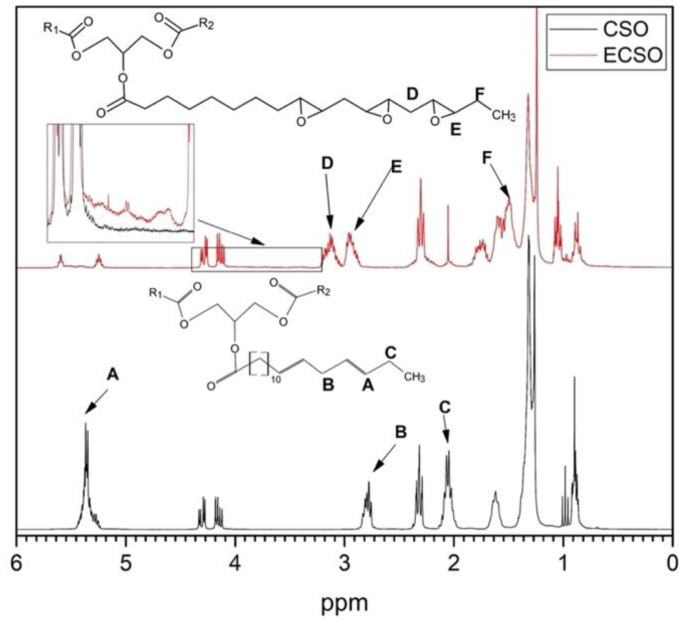
Comparison of CSO and ECSO by ^1^H NMR spectra.

**Figure 10 materials-15-03250-f010:**
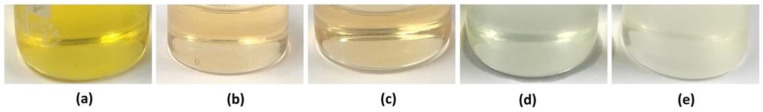
Influence of the MR and temperature on the color during the epoxidation process of the chia seed oil (CSO). (**a**) Untreated chia seed oil (CSO), (**b**) ECSO MR 0.75 at 60 °C, (**c**) ECSO MR 1.50 at 60 °C, (**d**) ECSO MR 1.50:1 at 70 °C and (**e**) ECSO MR 1.50 at 75 °C.

**Table 1 materials-15-03250-t001:** Iodine value and theoretical oxirane oxygen content of different vegetable oils.

Vegetable Oil	Initial Iodine Value (*IV*_0_, g I_2_/100 g)	Theoretical Oxirane Oxygen Content (*O_o_*, %)	References
Castor	84	5.03	[14]
Soybean	126	7.36	[27]
Rubber	156	8.95	[28]
Cottonseed	107	6.32	[11]
Linseed	188	10.6	[29]
Canola	112	6.60	[5]
Sunflower	130	7.57	[30]
Palm	62	3.76	[12]
Olive	127	7.41	[31]
Corn	115	6.76	[32]
Chia seed	197	11.05	Present study

**Table 2 materials-15-03250-t002:** Content of fatty acids presents in chia seeds (expressed as g of fatty acid/100 g of oil) and comparison with other studies.

	Fatty Acids (FAs)	In This STUDY	Demin et al. [53]	Kulczyński et al. [55]
**SFAs**	Myristic (C14:0)	0.06	0.04	0.06
	Palmitic (C16:0)	7.20	6.84	7.04
	Stearic (C18:0)	2.88	2.71	2.84
	Arachidic (C20:0)	0.55	0.28	0.02
	**Total SFAs**	**10.7**	**9.87**	**9.96**
**MUFAs**	Palmitoleic (C16:1)	0.09	0.24	0.03
	Oleic (C18:1)	4.32	6.17	7.3
	**Total MUFAs**	**4.41**	**6.41**	**7.33**
**PUFAs**	Linoleic (C18:2)	15.8	18.6	18.9
	γ-Linolenic (C18:3)	0.41	n.m ^1^	n.m ^1^
	α-Linolenic (C18:3)	68.6	64.4	63.8
	**Total PUFAs**	**84.9**	**83**	**82.7**

^1^ Where n.m means not mentioned.

**Table 3 materials-15-03250-t003:** Main parameters used to characterize the epoxidation process of the chia seed oil (CSO) at different molar ratios and temperatures.

Epoxidation	Temperature (°C)	MR ^1^	*IV_f_* ^2^	*X_IV_* ^3^	*O_o_* ^4^	*EEW*^5^ (g·eq^−1^)	*Y_OO_* ^6^	*S* ^7^
1	60	0.75	114 ± 0.94	42.2 ± 0.48	4.48 ± 0.11	357 ± 8.9	39.9 ± 1.02	0.946 ± 1.88
2	60	1.50	80.1 ± 0.87	59.4 ± 0.41	6.13 ± 0.12	260 ± 4.94	55.6 ± 1.07	0.935 ± 1.92
3	70	1.50	37.9 ± 0.62	80.8 ± 0.32	7.61 ± 0.10	210 ± 3.10	68.9 ± 0.98	0.853 ± 1.70
4	75	1.50	13.1 ± 1.52	93.4 ± 0.75	8.26 ± 0.11	193 ± 2.39	74.8 ± 0.78	0.801 ± 1.79

^1^ Hydrogen peroxide to double bond (H_2_O_2_:DB); ^2^ Final iodine value (g I_2_/100 g oil); ^3^ Conversion iodine value; ^4^ Oxirane oxygen content; ^5^ Epoxy equivalent weight; ^6^ Conversion to oxirane; ^7^ Selectivity.

**Table 4 materials-15-03250-t004:** Comparative of physico-chemical properties of chia seed oil (CSO) and epoxidized chia seed oil (ECSO).

Sample	Specific $\mathrm{Gravity}\;(\rho_{r})$	Dynamic Viscosity (mPa·s)	Colourimetric Coordinates			Colour Variation (∆E)
			**L***	**a***	**b***	
CSO	0.9285 ± 2.2·10^−3^	32 ± 0.45	75.4 ± 0.21	−6.27 ± 0.30	31.5 ± 0.35	0
ECSO MR 0.75 60 °C	0.9745 ± 1.2·10^−3^	109 ± 0.78	77.1 ± 0.28	6.73 ± 0.21	18.6 ± 0.21	18.4 ± 0.25
ECSO MR 1.50 60 °C	0.9901 ± 2.3·10^−3^	163 ± 0.73	76.5 ± 0.23	2.91 ± 0.17	13.8 ± 0.49	19.9 ± 0.19
ECSO MR 1.50 70 °C	1.0175 ± 1.7·10^−3^	420 ± 0.87	76.4 ± 0.28	−3.69 ± 0.35	4.47 ± 0.16	27.1 ± 0.44
ECSO MR 1.50 75 °C	1.0260 ± 2.9·10^−3^	558 ± 1.02	79.6 ± 0.42	−2.85 ± 0.36	4.50 ± 0.21	27.5 ± 0.42

## Data Availability

Not applicable.

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
