# Peer review of "Physicochemical Characterization of Novel Epoxidized Vegetable Oil from Chia Seed Oil"

_materials, 2022, doi:10.3390/ma15093250_

Round 1
Reviewer 1 Report
This research work assesses the development of a novel EVO obtained from chia seed oil, which might employed in various green products related to the polymeric industry. The content of this study is interesting, but there are some serious problems in data curation.
- Introduction: the author introduces the application and extraction methods of epoxidized vegetable oil, but neglects an important aspect, that is the reason for extracting epoxidized vegetable oil from Chia seed oil. Please explain further why the chia seed oil was chosen as the raw material.
- The author has previously published similar studies regarding epoxidized vegetable oil. Please clarify the similarities and differences in physicochemical characterization of epoxidized vegetable oil prepared in this study.
- In this research, all graphs and tables lack statistical analysis. Therefore, the repeatability of data cannot be determined, and the results obtained are not reliable.
Reviewer 2 Report
This paper proposes a study of the epoxidation of chia seed oid, which has a high iodine number. It is true that this study has not been performed, which is strange because chia seed oil has a lot of unsaturated groups.
They should speak about the use of this oil in the alimentary sector.
My other comments are below.
Introduction
+”One of the most promising renewables resources is vegetable oils (VO) because of their availability, relatively low cost, and non-toxicity” They should talk about the dilemma food versus fuel
+”…highlighting studies of linseed [9], cottonseed [10], soybean [11], karanja [12] or castor oil [13] [14], among others. …” the following references are missing
J.-L. Zheng et al., Kinetic modeling strategy for an exothermic multiphase reactor system: application to vegetable oils epoxidation by using Prileschajew method, AIChE Journal, 62(3) (2016) 726-741.
Cai, et al., Investigation of the physicochemical properties for vegetable oils and their epoxidized and carbonated derivatives, Journal of Chemical & Engineering Data, 63(5) (2018) 1524-1533.
+” This route presents some benefits compared to preformed peroxyacids, such as safer processing and handling, as well as requiring a minimum quantity of reagents to produce EVOs”
Actually, this route presents some risk of thermal runaway as mentioned in the article
S Leveneur, Thermal safety assessment through the concept of structure-reactivity: application to vegetable oils valorization, Organic Process Research & Development, 21(4) (2017) 543-550.
Nevertheless, this route if better than using oxygen.
+The study of ring opening has been studied by Cai et al., this article should be cited
Cai et al., Influence of ring opening reactions on the kinetics of bio-based cottonseed oil epoxidation, International Journal of Chemical Kinetics, 50(10) (2018) 726-741.
*Experimental section
+Did they measure the reaction temperature? Because this reaction is quite exothermic
+”… and stirring rate (220 rpm),…” did they vary the rotating speed to evaluate its influence on the kinetics?
+In Table 2, instead of Chia Seed Oil, they should put in this study.
+The kinetic study should be removed, with all respect due to the authors, this system is very complex and they proposed a very simplified approach.
This system is composed of several consecutive reaction steps, that cannot be expressed by Equation 11.
Equation 12 has a meaning for elementary reaction steps, which is not the case there.
The authors have enough results.
Conclusion
+“ The use of higher ratios of reactants and temperatures has been rejected due to the problems provided by sidereactions as the oxirane ring cleavage, which reduces the efficiency of the process” the term rejected should be changed.
Round 2
Reviewer 2 Report
They have replied to all questions and considered all suggestions.
In references, they should put all authors'names
